# Environmental Governance in Urban Watersheds: The Role of Civil Society Organizations in Mexico

Helena Cotler [1,*], Maria Luisa Cuevas [2], Rossana Landa [3] and Juan Manuel Frausto [3]

1   Centro de Investigación en Ciencias de Información Geoespacial, Mexico City 14240, Mexico
2   Environmental Consulting, Mexico City 03100, Mexico; marilucuevasfdz@gmail.com
3   Fondo Mexicano de Conservación de la Naturaleza, Mexico City 03900, Mexico;
    rossana.landa@fmcn.org (R.L.); juan.frausto@fmcn.org (J.M.F.)
*   Correspondence: hcotler@centrogeo.edu.mx

**Abstract:** Cities depend on several watersheds' ecosystems as the main source of ecosystem services for urban populations; however, this connection is not visible to decision-makers and citizens. The current governance structures do not contemplate the integrated management of the urban-rural territory by watershed; they establish few spaces for citizen participation, and limit the transparency of information. We use qualitative methods to analyze the work of the Civil Society Organization (CSO) in seven urbanized watersheds in Mexico, located under different socio-environmental conditions, to incorporate the watershed cities' management processes through new spaces of collaborative governance. Through environmental education campaigns, the CSOs raised awareness of the importance of watershed ecosystems to provide water for cities, explored the willingness to pay for their conservation, and the perception of the work of municipal water utilities. By promoting alliances between social sectors, the private sector, communities, and different levels of government, the CSOs built new institutions to increase the collaborative decisions and facilitate public participation, such as Watershed Committees, Citizen Observatories for Water and Consultative Councils. The incorporation of cities and citizens in the conservation of environmental services of the watershed was promoted through payment for environmental services programs. These processes of building new forms of governance are not linear. They depend on the convening and organizational capacity of the CSOs, the political will of the municipalities and states, as well as the socioeconomic conditions of citizens. In general, our results suggest that CSOs allow the formation of alliances that strengthen collaborations among stakeholders at different scales, increase government transparency and accountability, and provide a bridge of trust between upstream and downstream users in the watersheds.

**Keywords:** environmental governance; urban watershed; water urban planning; civic stewardship



## 1. Introduction

By 2050, more than 68% of the world's population will live in cities [1]. At global level, both the number of cities and that of their inhabitants have increased. After North America, Latin America is the second most urbanized region worldwide: 81% of the region´s population lives in cities [1]. This rapid growth in urbanization represents important challenges in terms of sustainability.

Urban expansion is considered one of the leading direct drivers of land use change and habitat loss, and is the most radical form of land transformation, deeply affecting biodiversity [2]. Urbanization alters the river´s natural flow regimes and water quality, ecosystem integrity and the sociocultural values of society, with a consequential loss of ecological services [3]. Other interrelated pressures, such as the loss or degradation of natural areas in watershed headwaters, soil sealing, spillage of contaminants, saline intrusion, deterioration in water quality, and the densification of built areas, pose additional challenges to the functionality of the ecosystem and, thus, for human well-being [4–6]. Currently,

worldwide watershed degradation costs cities USD $5.4 billion in water treatment annually, and many city governments realize the importance of investing in the sustainability of their watersheds [7].

Urban growth also alters neighboring ecosystems by increasing the demand for resources such as water, food and energy, and for generating large quantities of waste. It is estimated that cities require the ecosystem services provided by areas 500 to 1000 times larger than the cities themselves [8].

Diverse watersheds' ecosystems provide the primary ecosystem services for urban populations [9–11]. Traditionally, urban resilience has centered mainly on urban systems, such as the hydraulic infrastructure. In general, urban water planning and management are generally disconnected from planning and management at a watershed level [12]. However, urban-rural connections in the watersheds are critical for maintaining water supply [13]. In addition to the river flows, infiltration, recharge, sedimentation and pollution all depend on the land use present in the watersheds. As water links cities with their peri-urban/rural surroundings, watershed management is increasingly necessary to improve the resilience of urban water supplies [12,14].

A watershed is a biophysical unit of governance for water management in many countries, and a tool for hydrological modeling [15]. In recent years, studies regarding watershed governance have become relevant [16]. The approaches used and topics covered in those studies are wide ranging, considering, e.g., differences between urban and rural watersheds in different dimensions, highlighting sources of pollution, institutional complexity, and transaction and capital costs [17]. However, in most watersheds, these two conditions converge the urban and the rural, forming a complex geographic mosaic of physical, ecological, political and socio-economic diversity. In these territories, urban and rural processes and actors are interwoven with diffuse limits, for which reason only joint action can enable the provision and maintenance of ecosystem services.

In Latin America, the concern for activities of watershed management has historically been concentrated in rural areas. Mexico has broad watershed management experience in a rural context [18–20] emphasizing governance in the communities in order to improve the state of their ecosystems and productive activities. Currently, the main challenge is engaging the cities in the watershed management processes, which will require creating new spaces of collaborative governance [14].

In Mexico, water is a public good regulated by different levels of the government. Mexican cities face growing problems concerning water supply and sanitation problems, particularly for the low-income population [21]. In fact, municipal water utilities had limited technical and administrative capacity due to insufficient financial resources and the lack of skilled staff [22]. Currently, the watershed governance is fragmented between institutions of different governmental levels (federal, state and municipal) with jurisdictions and powers that are divided between the city and the rural part, ignoring their interactions. The performance of these institutions mismatches between ecosystem processes and management scales, for example, the rules, laws, policies, and formal and informal cultural norms, which govern the spatial and temporal extent of resource access rights and management responsibilities. Challenges related to temporality and scale can be seen as core governance dilemmas [23].

Mexico, similar to other countries in Latin America, is home to high biodiversity [24] and is almost the most unequal region of the world in terms of economic income [25] and land distribution [26], which generates strongly polarized social structures. In addition, extractive development has intensified over the preceding decades, resulting in deep socio-environmental conflicts, especially in territories that are home to indigenous people [27,28]. As a result, Mexico, as well as Colombia, Honduras, Brazil, and the Philippines, have had the highest homicide rate of environmental advocates and land defenders in recent years [29].

Severe effects of climate change, e.g., drought, heavy rain, flooding, and pest, stress watershed ecosystems and make watershed management more critical in the country.

In these territories, the absence of the state leaves open niches that can be occupied by CSOs [30]. The analysis of the role of CSOs in different areas has been present since 2000 [16]. Several authors have analyzed the role of CSOs in specific themes, such as agriculture [31,32], or their role in municipalities [33]. Other research has focused on evaluating the role of CSOs indirectly in political outcomes such as governance [34–37]. However, participation by CSOs in the political sphere has been the object of broad discussion in the academic milieu, especially because there has been an additional growth in the number of these organizations and the list of their types and functions has lengthened as well [30]. In this sense, [38] identify some CSOs' limitations to create an effective long-term partnership, because partnerships between the state and CSOs may distort the very nature of these organizations as representative of interests of the society. Besides, the main reason for CSOs to exist is to do with the belief that they act for collective objectives, which may occasionally be excluded from ordinary public policy formulation procedures [30].

Although these studies explored different aspects of CSOs' effects in the political sphere, very little research has measured the feedback effects of CSOs in urban watershed governance. On this subject, we would like to contribute with our research questions.

Accordingly, the objective of this study is to analyze the experience in seven urban watersheds in order to document the (i) mechanisms used by Civil Society Organizations (CSOs) to forge links between cities and watershed management, (ii) the construction of governance within the watershed and the urban water management, and (iii) the mechanism to encourage the participation of the cities in hydrological services conservation.

## 2. The Role of Civil Society Organizations in Urban Environmental Governance

In many cities, Civil Society Organizations (In the context of this article, we refer to Civil Society Organizations (CSO) as all local groups related to the local environmental stewardship) have focused on the conservation, restoration, administration, monitoring and defense of natural richness, as well as educating the public on a wide range of topics related to the maintenance of the local environment. They also form a crucial component of the structure of urban environmental governance by establishing networks with other local groups and governmental agencies [39,40]. These organizations fulfill a particular role since they can provide access to grant funds, scientific research, technical support, and trained personnel with which to implement restoration projects [17]. Moreover, they have the potential to compile additional information, obtain new perspectives of problems and develop more creative solutions, which could increase the legitimacy of decision-making, leading to a more significant appropriation of the resulting decisions, less litigation and better activities implementation [41].

Diverse experience shows that it is necessary to develop and strengthen collaborations among stakeholders at different scales to improve water management in the territory [42]. In the cities, cooperation between the institutions responsible for the managing of watersheds and the water services administrators continues to be an important challenge for the sustainable management of urban water resources [42].

The role of CSO as a key actor, providing bridges between rural and urban environments, has been widely studied and recognized, particularly in terms of their critical role in achieving a balance among sectors of government, businesses, and civil society [39]. This bridging function can improve ecosystem management, influencing the quantity and quality of ecosystem services for urban areas [43]. It can also counteract technocratic culture, short-term political cycles or the human tendency to resist change, which constitutes some of the main barriers to non-traditional water management approaches, including blue-green infrastructure [44].

The experience of local environmental groups (CSO) working together with governmental agencies and the private sector strengthens watershed administration [45], as well as fortifying social participation and increasing government transparency and accountability, which is essential to the achievement of adaptive watershed management [41]. From this governance, it is possible to achieve sustainable development goals (SDG) directed

towards the cities (https://www.un.org/sustainabledevelopment/es/cities/ (accessed on 18 August 2020) that recognize the need to support positive economic, social and environmental links among urban, peri-urban and rural zones, strengthening regional development planning (SDG 11.A), as well as increasing the capacity for planning and participative management (SDG 11.3).

The Watersheds and Cities Program of the Fondo Mexicano para la Conservación de la Naturaleza (FMCN) funds local CSOs to protect and recover the watersheds that supply water to the cities. It is achieved through capacity building, technical assistance, funding and inter-institutional synergies (https://fmcn.org/es/programas-proyectos/bosques-y-cuencas (accessed on 17 July 2020). Since 2001, the program has been applied in 12 watersheds of medium-sized cities in Mexico.

The FMCN program establishes an alliance with a local partner, in each selected watershed, leading watershed management through planning, resources investment, and institutional collaboration. This social leadership results from local support, experience in nature conservation, multi-actor work and linkages with local stakeholders. It has created decision-making spaces for watershed management and provision of water, reaching beyond administrative limits and electoral cycles.

## 3. Methodology

This research is based on the knowledge, experience and actions carried out by the CSOs and their members. To capture these opinions and the lessons learned we use a qualitative analysis.

(i) Case studies

This study selected seven CSO-watersheds initiatives, comprising the cases in which experience has been developed to construct a watershed-city relationship. The watersheds are: La Paz watershed (South Baja California), Baluarte and Presidio rivers (Sinaloa), Bravo-San Juan and Sierra Madre Oriental (Coahuila), Laja river (Guanajuato), Pixquiac (Veracruz), Coatzacoalcos and Minantitlán (Veracruz) and Valle del Jovel (Chiapas). The distribution of the watersheds where CSOs work covers different climatic zones of Mexico, from the hot arid zone in the north (1 watershed), arid zone (1 watershed) to humid subtropical zone in the center and south (3 watersheds) and the tropical zone in the southeast (2 watersheds) (Figure 1). All the CSOs have a local scope, and clear principles that oriented their activities, e.g., "pal", "protect and restore the hydrological and ecological functions", "rural development with justice", "linking cities users of the water supply", "establishing strategic alliances and participating in networks with academia, government, private initiative and civil associations for the conservation and sustainable management of Mexico's biodiversity".

(ii) Data collection process

The information for this study comes from various qualitative sources: (i) semi-structured interviews with key actors of the CSOs, addressing themes relating to their experience of watershed management, identification of the main problems of the watershed, and the actions carried out and strategies adopted to incorporate the cities into watershed management; (ii) internal documents of each CSO (work programs, management plans and actions) to understand the objectives and their socio-environmental diagnosis, (iii) the last external review of the program to know their perspective about the performance of each CSO, and (iv) field trips to each watershed, where actions were reviewed and informal interviews held with local actors. The size and maturity of each of the CSOs are different. Some have their own interdisciplinary teams, while others are supported by alliances with other organizations. Depending on the size and complexity of each CSO, the number of people interviewed was determined, seeking to cover different profiles within each CSO, to complement the information and obtain different perspectives (Table 1).

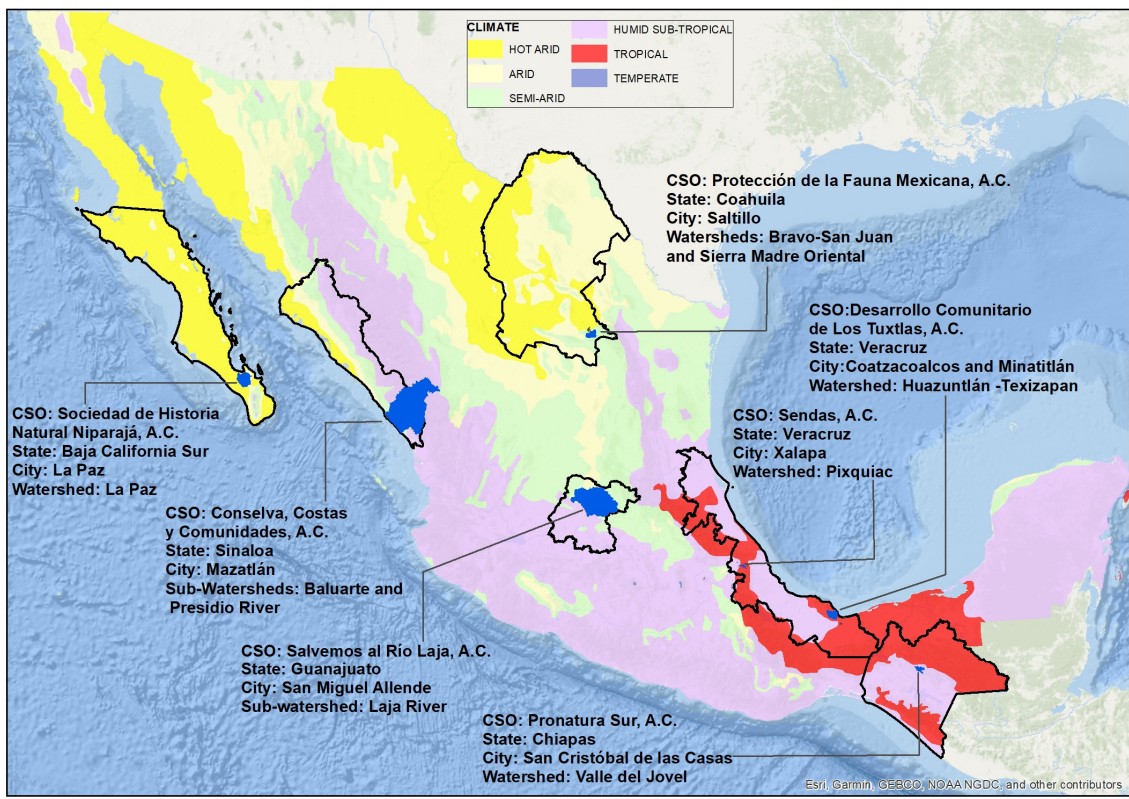

**Figure 1.** Location of the Watersheds and their Respective Climate Zones.

(iii)   Data analysis process

The semi-structured interviews were translated and subjected to an analytical framework approach, which is useful to describe important processes, such as the history of the CSO and chronology of their actions. This analytical framework follows a case study approach that allows organizing the data by each specific CSO.

The inductive analysis allowed us to discover patterns and categories through developing a coding scheme for each case study. In our study, the coding scheme is (1) Hydrologic characteristics and socio-environmental problems of the watershed, (2) Main actions to increase governance in the watershed (surveys, campaigns, alliances); (3) Track record and relationships with government agencies to improve the governance of the watershed; (4) Creation of new governance instruments.

Content analysis allowed us to analyze the core content of the interviews and the documents. Subsequently, a comparative analysis of the priorities of each CSO in each watershed was carried out [46]. The analysis of this information made it possible to identify the main socio-environmental problems of the watershed and their cities; generate a narrative on the construction of governance with other watersheds' governmental and social actors, and identify the main challenges that face the CSOs. The analysis of qualitative data is very useful to capture the perceptions of different actors, build a narrative over time, appreciate the tensions and evaluate the results, which are phases of governance. For this reason, this study method has been used by several authors at the national and international level [2,17,40,47].

Throughout interviews and document review, we followed the Ethics Code of the Association of Social Anthropologists of the UK and the Commonwealth. Ethical Guidelines for Good Research Practice.

**Table 1.** Location and characteristics of cities and watersheds-case studies. 1: Main sources: interviews and FMCN [48] * These cities are located outside the limits of the hydrographic watersheds but their water supply depends mainly on the watershed studied.

| City and Watershed | Civil Society Organization | CSO Mission | Person Interviewed and Position within the Organization | Title and Number of Documents Consulted | Type of Actions Reviewed in the Visits | Main Problems in the Watershed/Sub-Watershed |
|---|---|---|---|---|---|---|
| Mazatlán, Sinaloa *Watersheds Baluarte and Presidio* | CSO Conselva, Costas y Comunidades, A.C. https://www.conselva.org/ (accessed on 23 July 2020) | The organization works to build a prosperous and sustainable future for the inhabitants of northwestern Mexico, articulating society, communities, companies, academia and government in the sustainable management of watersheds, water and biodiversity. | 1. Executive Director 2. Coordinator of the Biodiversity and Watersheds Program, 3. Operational Technician, 4. Coordinator of the Program for Strengthening Social Capital and Sustainable Production | 1. [49] 2. CONSELVA https://www.conselva.org/post/por-que-cada-vez-hay-menos-agua (accessed on 23 July 2020) 3. CONSELVA https://www.conselva.org/post/la-cuenca-el-origen-del-agua (accessed on 23 July 2020) 4. CONSELVA https://www.conselva.org/post/unidos-por-la-cuenca (accessed on 23 July 2020) | 1. Soil erosion monitoring plots 2. Runoff plots 3. Automatic monitoring (Coshocton wheel) of runoff, flow and sediments area 4. Soil conservation works | Agriculture exerts intense pressure on water availability, causing decreased piezometric levels. Livestock on slopes causes soil erosion |
| San Miguel de Allende, Guanajuato *Sub-watershed Laja river* | CSO Salvemos al Río Laja, A.C. https://agua.org.mx/biblioteca/salvemos-al-rio-laja-ac/ (accessed on 24 July 2020) | The organization protects and restore the hydrological and ecological functions in the micro-watersheds, to ensure in the long term the environmental health and the recharge capacity of the aquifer and guarantee the supply of quality water to the inhabitants of urban areas and rural communities in San Miguel de Allende | 1. Executive Director 2. Forestry technician | 1. [44] 2. [50] | 1. Soil conservation works in the upper watershed 2. Gabion weirs 3. Gully treatment measures 4. Backyard farming 5. Rainwater harvesting systems. | Desertification of livestock production soils; presence (natural) of fluoride and arsenic in water wells, high water consumption in industrial and urban centers Changes in rainfall patterns |
| Xalapa, Veracruz *Watershed Pixquiac* | CSO Sendas, A.C. https://sendas99.wordpress.com/que-hacemos/gestion-integral-de-la-cuenca-del-rio-pixquiac/ (accessed on 27 July 2020) | The organization seeks to promote sustainability, through the good management of natural resources, as well as rural development with justice and search for a new environmental rationale that allows a good life for people in the countryside and the city | 1. Cordinator of the Pixquiac river basin project 2. Co-coordinator of the Pixquiac river basin project | 1. [20] | 1. Permanent soil monitoring plots 2. Monitoring of silvopastoral modules 3. Forest conservation area | Deforestation, erosion of soils due to livestock production; urban growth; crops that present a high consumption of water and pesticides (e.g., potato) |

**Table 1.** *Cont.*

| City and Watershed | Civil Society Organization | CSO Mission | Person Interviewed and Position within the Organization | Title and Number of Documents Consulted | Type of Actions Reviewed in the Visits | Main Problems in the Watershed/Sub-Watershed |
|---|---|---|---|---|---|---|
| La Paz, Baja California Sur *Watershed La Paz* | CSO Sociedad de Historia Natural Niparajá, A.C. http://niparaja.org/ (accessed on 28 July 2020) | The organization promotes the conservation of habitats, natural resources and priority ecological processes in Baja California Sur | 1. Coordinator of the Program Water and City 2. Collaborator of the Program Water and City 3. Collaborator of the Program Water and City) | 1. [45] 2. Niparajá https://niparaja.org/wp-content/uploads/2015/06/Caso_SAA_TF_L.pdf (accessed on 28 July 2020) 3. Niparajá. https://niparaja.org/wp-content/uploads/2015/06/Monitoreo-calidad-agua-La-Paz-Carrizal-As-y-Fe-con-3er-muestra-pu%CC%81blico.pdf (accessed on 28 July 2020) 4. Niparajá Modelación numérica para la determinación de flujos subterráneos. Sitio Piloto: La Paz, Baja California Sur, México | 1. Urban green infrastructure 2. Soil and vegetation conservation measures | Intrusion of saline water into the aquifer. Streams and rivers polluted with solid residues; lack of transparency and accountability in quality and quantity of water |
| Coatzacoalcos, Minatitlán, Veracruz *Watersheds Huazuntlán Texizapan* * | CSO Desarrollo Comunitario de Los Tuxtlas, A.C. https://agua.org.mx/wp-content/uploads/2011/09/historia_de_conservacion_decotux.pdf (accessed on 31 July 2020) | The organization is dedicated to facilitating environmental co-management processes for sustainable development and ecological restoration in indigenous communities of the Sierra de Santa Marta and Los Tuxtlas, in a way articulated with strategies for prevention, mitigation and adaptation to climate change, linking cities users of the water supply in the South of Veracruz, with the potential for replication in other hydric regions. | 1. President and General Coordinator 2. Business associate | 1. [51] | No visited | Deforestation; sedimentation of the dams that supply the cities; saline intrusion into wells and contamination by the industrial zone; deficient hydraulic infrastructure; climatic variability |

Table 1. *Cont.*

| City and Watershed | Civil Society Organization | CSO Mission | Person Interviewed and Position within the Organization | Title and Number of Documents Consulted | Type of Actions Reviewed in the Visits | Main Problems in the Watershed/Sub-Watershed |
|---|---|---|---|---|---|---|
| Saltillo, Coahuila *Watersheds Bravo-San Juan Sierra Madre Oriental* | CSO Protección de la Fauna Mexicana, A.C. http://profauna.org.mx/ (accessed on 5 August 2020) | The organization's objectives are to promote conservation of priority species and ecosystems under various management schemes aimed at their sustainable use and recovery, generate strategies and innovative tools for their application in the conservation of biodiversity and ecosystems in Mexico, strengthen the capabilities of key actors through the training of human capital that contributes to conserving priority species and ecosystems in Mexico and establishing strategic alliances and participating in networks with academia, government, private initiative and civil associations for the conservation and sustainable management of Mexico's biodiversity. | 1. Director 2. Zapalinamé project operational coordinator 3. Coordinator of Fire management program, soil and water conservation and forest resources project 4. Technician 5. Livestock management and land acquisition and conservation 6. Technician in charge of social participation and landscape connectivity 7. Technician in charge of environmental interpretation and recreation 8. Coordinator of environmental culture and community development 9. Surveillance and contamination 10. Quality of life and community development, plant production and forest health 11. Promotion of research and monitoring of birds) | 1. [52] | 1. Birds monitoring 2. Forest health monitoring 3. Monitoring of burned areas 4. Soil conservation works | Scarcity of water, destruction of wells, prolonged drought, and inadequate hydraulic infrastructure |
| San Cristóbal de las Casas *Watershed Valle de Jovel* | CSO Pronatura Sur, A.C. http://www.cuencavalledejovel.org/ (accessed on 12 August 2020) | The organization seeks to develop conservation models that promote alternatives for the use and management of natural resources that benefit communities. For this reason, we work promoting the participation of society and hand in hand with communities, organizations and owners. | 1. Operational Coordinator 2. Deputy Director of Conservation) | 1. [19] | 1. Private natural protected areas 2. Educational Center 3. Soil conservation works in agricultural plots | Deforestation; scarcity and contamination of water; health problems due to contamination of water sources |

## 4. Results

The cities supplied by the watersheds that we analyzed present populations that range from 160,000 to 725,000 inhabitants. These cities are all located between the middle and the lower part of their watershed; therefore, the management of the upper part of each watershed is crucial to maintain the provision of ecosystem services (Table 2).

The sources of water vary across the watersheds. In the hot arid, arid, and humid subtropical climate zones, the supply comes from groundwater, while the watersheds in tropical climatic zones are supplied by surface water. In all watersheds, the surface and subterranean water in cities face three main threats: deterioration of the ecosystems that provide these hydrological services, the intense consumption of water and its contamination as a result of different activities such as agriculture, livestock production and industry, and the lack of adequate water treatment facilities. The contribution from the watershed to the city´s water supply varies from 38% (The remaining percentage of the water supply of the cities comes from surrounding watersheds through water transfers.) to 100%. In four cities the contribution of their watersheds is greater than 90% [48].

Each CSO analyzed their respective watershed hydrological service and acknowledged the anthropic impacts on its quality and quantity. Through forums and meetings, they determined that water was a critical theme, and this confluence allowed many urban and rural sectors to share critical ideas and challenges. In this way, their interest converged with the FMCN program.

The FMCN has been funding CSOs for over six years (from seven to 19 years). However, these CSOs have more than 20 years, of operating in the watersheds undertaking different tasks in the rural areas. Their work includes the coordination of participative planning processes, working with rural communities for productive improvement, definition and participation in the management of protected natural areas, protection of wetlands and the biodiversity of the watersheds, generation of technical documents, building of alliances and lobbying of governmental institutions in order to prevent mining and highway construction activities from destroying ecosystems.

These CSOs have a strong presence in the territory, and have built trust and partnerships among the population.

### 4.1. Mechanisms for Incorporating Cities into Watershed Management

As stated above, the CSOs had vast experience of working mainly in the rural areas of the watersheds. To incorporate the cities into watershed management, the CSOs in coordination with the FMCN program established two primary mechanisms:

(i)　They delimited the hydrographic unit in which each city is located. Cities traditionally are managed within their administrative entities (municipality and state), rarely approached in hydrographic units; therefore, the delimitation of the city within the watershed limits was a novel step.

This delimitation followed hydrological patterns adapted to each situation. For example, when the cities cross hydrographic units, the sub-watershed or watershed of the city was considered (i.e., Coatzacoalcos-Minantitlán, Veracruz). In other cases, the limits of the protected natural areas were taken as the watershed limit because of their importance for groundwater recharge (Saltillo, Coahuila). A third case was that of the sub-watersheds that were grouped together to cover the dams and the rivers that supply the city (Mazatlán, Sinaloa). In the latter case, the water is transferred between watersheds.

(ii)　Before developing plans and implementing actions to incorporate the cities, the CSOs conducted surveys to determine the perceptions and knowledge of the urban population regarding their sources of water supply, its quality and threats, and the roles of the watershed´s ecosystems in the provision of water. Understanding the sociocultural perceptions of the relationships between the urban population and nature was essential to promote collective responses for the sustainable management of ecosystems. These surveys were conducted with the participation of local universities.

Table 2. Location and characteristics of cities and watersheds—case studies. 1: Main sources: interviews and FMCN [48] * These cities are located outside the limits of the hydrographic watersheds but their water supply depends mainly on the watershed studied.

| Watershed | City | Total Population in the Watershed/Sub-Watershed | Total Population in the City (2010) and Percentage of the Population of the Watershed | Location of the City within the Watershed | Primary Source of Water Supply in the Cities | Contribution of the Watershed to the the City's Water Supply | Actions | Results |
|---|---|---|---|---|---|---|---|---|
| Presidio and Baluarte river | Mazatlán | Baluarte river: 60,584 inhab. Presidio river: 494,035 inhab. | 381,583 inhab. (77.24% of the population of the watershed Presidio) | Lower | Surface water, stored in dams | 100% | An initial survey was conducted with another survey two years later. These surveys addressed themes relating to the knowledge of water and its relationship with the watershed. | In two years, understanding of the relationship of water with the watershed increased from 5 to 20% |
| | | | | | | | Three information/communication campaigns were conducted. Their objectives were first to explain the hydrological cycle (campaign: "Nuestra Agua" (Our water)). Second, the role of the watershed in the provision of water was explained (campaign: "El agua de tu casa viene de la cuenca" (The water in your house comes from the watershed.)) Finally, the actions that must be undertaken to maintain the water in the future were addressed by the Campaign: "Nuestro futuro" (Our future) | Different messages were presented according to the audience (industrial, agricultural/livestock production, domestic, governmental) |

**Table 2.** *Cont.*

| Watershed | City | Total Population in the Watershed/Sub-Watershed | Total Population in the City (2010) and Percentage of the Population of the Watershed | Location of the City within the Watershed | Primary Source of Water Supply in the Cities | Contribution of the Watershed to the the City's Water Supply | Actions | Results |
|---|---|---|---|---|---|---|---|---|
| Sierra de Zapali-namé protected natural area | Saltillo | 1484 inhab. | 709,671 inhab.* | Middle | Aquifer | 40% | A survey was conducted to identify the knowledge of the population regarding the origin of their source of water and to determine whether they were willing to donate to preserve these sources. | This study determined that the people did not know where their water came from, but there has a willingness to donate to the conservation of their supply source (80% acceptance). |
| | | | | | | | Communication campaign "Una razón de peso", which lasted four years. A subsequent fundraising campaign "Un peso se va como agua, haz que tome cauce" (A peso goes away like water, make it flow) was conducted. | Through the body that operates the water supply, the CSOs collected donations from 47 thousand families out of a total of 205 thousand users of water. |
| | | | | | | | Radio programs twice weekly, publicity in buses, a gazette named "*Pregonero de Zapalinamé*", a quarterly publication with a circulation of 40,000. Annual surveys of 1000 people are conducted from the city and countryside. | The number of donors and the amount donated have both increased, as the awareness of the origin of the water supply and the importance of conserving the natural resources that allow recharge of the aquifers. There is clear recognition that the water of the city comes from the Sierra de Zapalinamé. |

**Table 2.** *Cont.*

| Watershed | City | Total Population in the Watershed/Sub-Watershed | Total Population in the City (2010) and Percentage of the Population of the Watershed | Location of the City within the Watershed | Primary Source of Water Supply in the Cities | Contribution of the Watershed to the the City's Water Supply | Actions | Results |
|---|---|---|---|---|---|---|---|---|
| Río Laja | San Miguel de Allende | 599,754 inhab. | 69,811 inhab. (11.64% of the population of the sub-watershed) | Middle | Aquifer | 99% | Survey of the urban population to determine their disposition towards paying for the conservation of the recharge zones (payment for environmental services). | The willingness of the population to contribute economically to support projects of restoration, as long as the budget is not managed by a government agency |
| Pixquiac river | Xalapa | 7537 inhab. | 424,755 inhab.* | Middle | Surface water | 38.5% | Survey of the urban population (domestic use) with a socio-economically stratified sample to identify the knowledge regarding the origin of the water and to determine their disposition towards contributing economically to the protection of water sources. | Only 4% of the 120,000 people surveyed reported knowledge regarding where their water came from. A total of 50% of those surveyed were willing to make donations of between $0.25 and 0.5 USD per month. |
| Texizapan and Huazuntlán river | Coatzacoalcos and Minatitlán | 569,054 inhab. | 235,983 inhab.* | Lower and middle | Surface water | Coatzacoalcos: 80% Minatitlán: 40% | A watershed tour was implemented, consisting of an excursion through the high part of the watershed, so that people could learn from where their water comes from and the origin of the rivers. This tour is open to the population and is conducted in coordination with the communities and *ejidos.* Environmental education presentations in schools, using models and partners from different organizations. | Increased awareness among both the urban and rural populations regarding the need for watershed conservation. |

**Table 2.** *Cont.*

| Watershed | City | Total Population in the Watershed/Sub-Watershed | Total Population in the City (2010) and Percentage of the Population of the Watershed | Location of the City within the Watershed | Primary Source of Water Supply in the Cities | Contribution of the Watershed to the the City's Water Supply | Actions | Results |
|---|---|---|---|---|---|---|---|---|
| La Paz | La Paz | 272,711 inhab. | 215,178 inhab. (95% of the population of the watershed) | Lower | Aquifer | 90% | A survey exploring the perception, consumption service, and use of water and its quality. A campaign "*el agua no viene de la llave, viene de la sierra*" (*The water does not come from the faucet, it comes from the mountains*) was launched on radio and television. | Among the population, there is widespread ignorance regarding the origins of their water supply, perhaps as a result of the high proportion of immigrants in the population of La Paz (around 40%) |
| | | | | | | | Permanent workshops for raising awareness are held at elementary schools and universities, to raise awareness about the importance of the mountains, as the main catchment area, and about water problems in the region. | Students in 28 schools have been trained. Alliances have been created with the Mexican National Council of Science and Technology |

In the city of Mazatlán, the CSO conducted two surveys (2015 and 2017) (Figure 2), including two information campaigns to explain the hydrological cycle "Our water" ("*Nuestra agua*") and to explain the role of the watershed in the provision of water "The water in your house comes from the watershed" ("*El agua de tu casa viene de la Cuenca*").

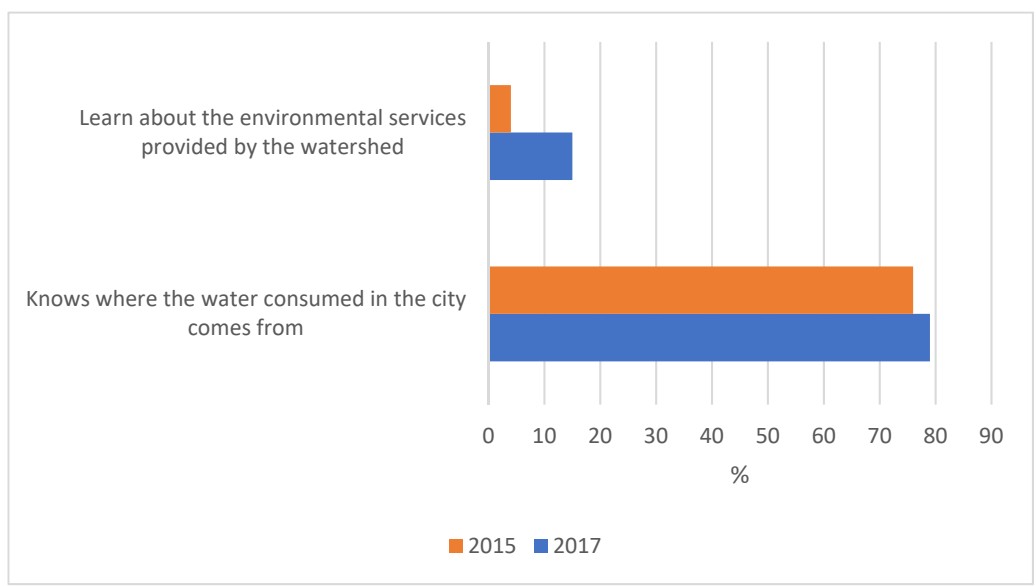

**Figure 2.** Survey results in the City of Mazatlán [49].

Most of the urban population knows the origin of their water but still does not relate it as an environmental service that comes from the ecosystems in the watershed. This knowledge increased slightly as a result of the information campaigns.

The campaigns also sought to inform the population about those responsible for water management in the city: the Municipal Water Utility. Understanding among respondents rose from 64% to 95%. Respondents also indicated the need for the Municipal Water Utility to make its resources transparent (from 72% to 90% between 2015 and 2017).

In the City of Xalapa, a survey was implemented in 2015, as part of a governance construction path (Figure 3) and the campaign "Water comes from the watershed" ("*El agua viene de la cuenca*"). From the responses obtained, we observe that citizens were not aware of the origin of their water. However, they could establish the relationship between the forests in their watersheds and the water they consume.

In the City of La Paz, the CSO conducted two surveys (2012 and 2018) (Figure 4) to analyze the citizens´ knowledge regarding the sources of water, its management, and its quality. The 2012 survey evidenced that 59% of those surveyed do not recognize that they live in a semi-desert area, and 57.9% do not perceive water scarcity as a problem.

Between the two surveys, the CSO carried out several information campaigns such as, "Water does not come from the tap, it comes from the mountains" ("*el agua no viene de la llave, viene de la sierra*") to sensitize and inform the urban population about the water issue. The results show an increase in the understanding of the informed urban population about the origin of their water (It is important to highlight that surveys were conducted with different methodologies and the CSO recognizes the data may present biases).

Conversely, 68% of the population of La Paz, was not aware which recharge areas are for underground water, so they do not relate urban water to its watershed [45].

The CSOs have conducted several education and communication campaigns in most cities using various media sources (public information campaigns, radio, TV, bulletins, infographics, press conferences, fairs, and exhibitions). These campaigns have emphasized the importance of the health of the watershed to the water sustainability of the cities.

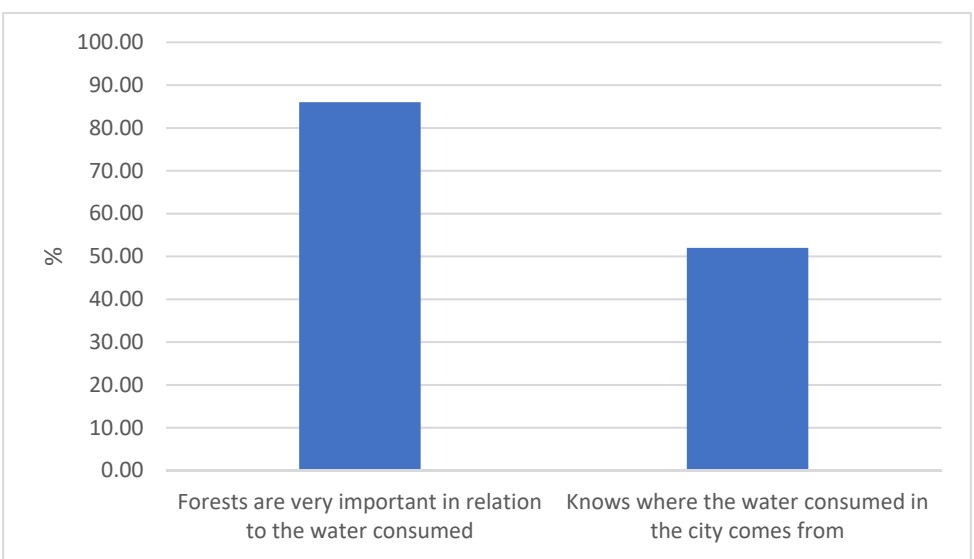

**Figure 3.** Survey Results in the City of Xalapa [53].

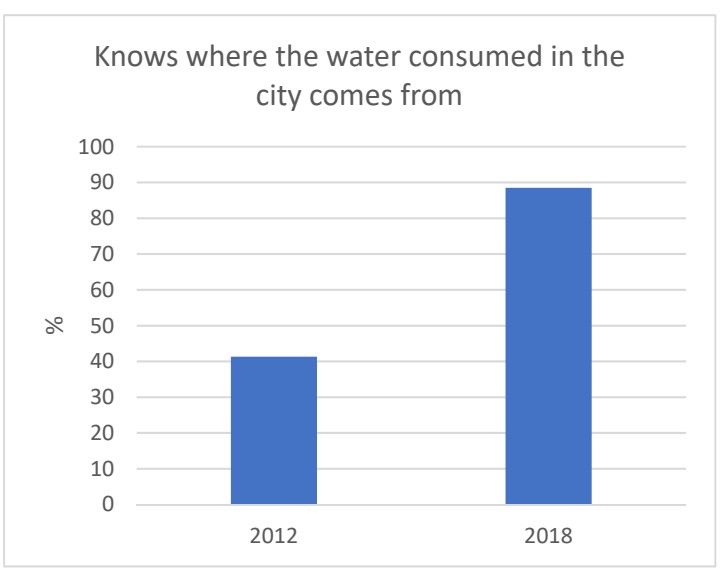

**Figure 4.** Survey Results in the City of La Paz [45].

*4.2. Construction of Governance in the Watershed and Water Management*

In the watersheds and the cities, the complexities of actors and political entities are multiple and diverse, which raises the transaction costs of establishing strategic plans, sharing information, and management efforts.

Currently, governance in the watersheds is fragmented between institutions of different governmental levels (federal, state, municipal), with jurisdictions and powers that are divided between the city and the rural part, ignoring their interactions.

Besides governmental institutions, it is apparent that the private sector, local environmental groups, and universities also influence the plans and programs that modify the watershed and its city (Figure 5).

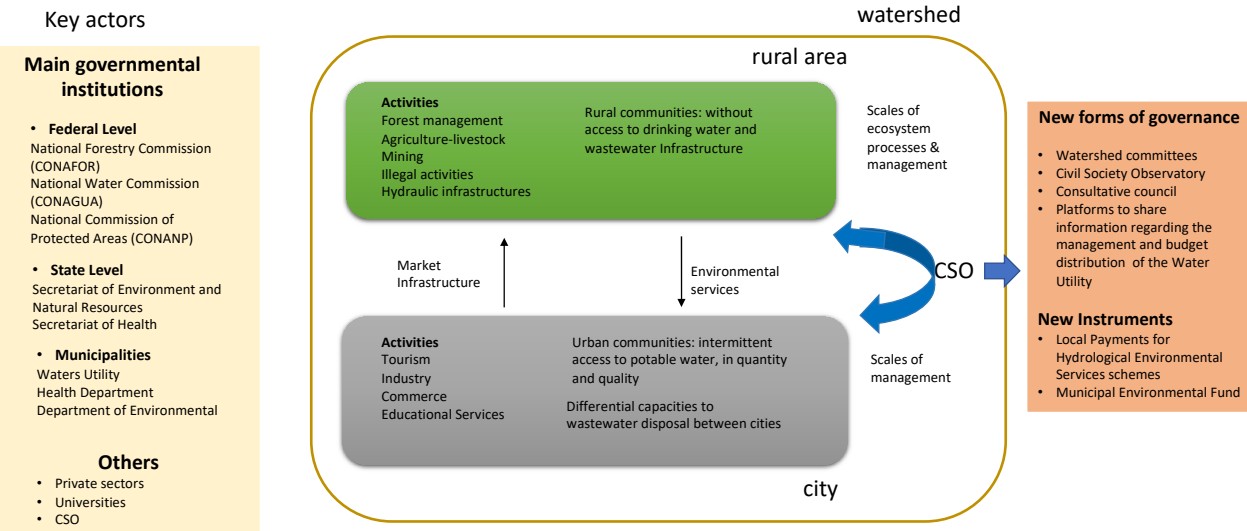

**Figure 5.** Water Governance in the Watershed, Formally and Alternatively Driven by CSOs.

Citizen participation in water decision-making is established through top-down institutions (Watershed Councils), with little representativeness and legitimacy [54,55]. Within them and in limited territories, CSOs can form watershed committees. However, these figures are not binding with the Watershed Councils, as they do not have resources, and their operating plans depend on the political will of government representatives (CONAGUA).

To increase the collaborative decisions and facilitate public participation, the CSOs created and strengthened agencies and instruments of governance, such as Watershed Committees, Citizen Observatories for Water and Consultative Councils, and instruments that promote more active participation of the cities in watershed management. Below, we exemplify the cases of CSOs that work in the cities of Xalapa, Coatzacoalcos, and San Cristobal de las Casas to construct local Watershed Committees and the CSOs that promote transparency in water utilities in La Paz and San Miguel de Allende.

(a)    Construction of Local Watershed Committees

It was necessary to create institutions with local representation and a territorial approach to conserve ecosystems and avoid infrastructure and activities that deteriorate them.

### 4.2.1. City of Xalapa (Veracruz)

In the city of Xalapa, the common interest in preserving natural resources prompted the preparation of a participatory diagnosis of the watershed's problems, which led to the creation of the Pixquiac Watershed Committee (Cocupix) in 2006. This citizen platform integrated diverse stakeholders, inhabitants of the sub-basin, private landowners, ejidos, producers, CSOs, and academics. From Cocupix, the CSO sought dialogue with government agencies of the City of Xalapa, as a beneficiary of environmental services. Ten years' work made it possible to create, together with the city council and the municipal water utility, the Environmental Services Compensation Program (PROSAPIX), an instrument that uses the city's budget to the conserve and restore areas that provide environmental services was supported. In addition, investments have been made in productive projects to reduce pressure on natural resources, generate real economic alternatives, and encourage inhabitants to stay in their communities. Rural participation is currently low due to clientelism relations, typical of local idiosyncrasy. The steps followed by the CSO to institutionalize the payment for environmental services program are explained below.

### 4.2.2. Cities of Coatzacoalcos-Minantitlán (Veracruz)

The watersheds of Coatzacoalcos-Minantitlán (Veracruz) suffered intense deforestation in the 1960s and 1970s, promoted mainly by government programs. This condition in-

creased their vulnerability to hurricanes and torrential rains. Hurricane Stan in 2006 caused landslides in large areas and sedimentation of dams that feed the cities of Coatzacoalcos and Minantitlán. This event triggered organizational processes among the communities (Figure 6), forming watershed committees, where Popoluca, Nahua and mestizo indigenous people intervened. The indigenous communities in the upper watershed are among the most marginalized in the country. Their territories, which constitute their livelihood, are seen from the urban environment as spaces to be conserved (decotux.org/la-importancia-de-la-información). Facing the disasters of Hurricane Stan led to the elaboration of a management plan among 11 communities, which was later expanded to 35, in the watershed subcommittee of the Texizapan-Huazuntlán river. Actions are tailored to each condition, from payments for environmental services with federal and state budgets, to soil restoration actions, and various studies to better understand the impact and possibilities for adaptation to climate change in the region (www.decotux.org (accessed on 25 August 2020). The limited participation of the municipal government shows their lack of interest in the conservation of the upper watershed.

| 2006 | 2006-2008 | 2009 | 2015-2021 |
|---|---|---|---|
| Storm Stan causes landslides and silts in dams that supply the cities of Coatzacoalcos and Minatitlán.<br><br>State government grants budget for recovery actions | Organization of 11 communities to prepare a Watershed Management Plan, for water availability | An agreement with the municipalities of Coatzacoalcos and Minatitlán was attempted, to allocate resources for restoration projects, with no favorable response.<br><br>Creation of the Huazuntlán River Basin Subcommittee, to promote citizen participation | Watershed Subcommittee, representing 35 communities, is part of the Coatzacoalcos River Basin Council (led by CONAGUA).<br><br>Restoration and production actions are intensified, directed by a management plan prepared by the communities, and financed by the Federal Government. |

**Figure 6.** Temporary Process for the Construction of the Watershed Committee of the Huazuntlán River (2006–2021).

### 4.2.3. City of San Cristobal de las Casas (Chiapas)

In San Cristobal de las Casas, Chiapas, a participatory process was developed to elaborate a management plan for the Jovel Valley Watershed Committee (2014) (http://www.cuencavalledejovel.org/ (accessed on 16 July 2020). This Committee included representatives of communities, academic institutions, businesses, and government. The limited political will of government representatives obstructed the implementation of management programs and the provision of financial resources. A CSOs alliance was formed to monitor water quality in the San Cristobal de las Casas distribution network. Based on these results, corrective actions were requested including the extension of the sampling network. This request received no response from the municipal government or the Municipal Water Utility. Faced with the problem of drying up the springs and streams that used to provide water to peri-urban communities, the CSO has developed projects to provide families with cisterns and water systems.

Given the lack of government funding, over the last few years the CSO has conducted several legal and feasibility studies on financing mechanisms for the sustainability of the Committee's actions, evaluating issues such as adding a fee to the potable water bill, obtaining a percentage of the lodging tax for watershed actions, or establishing a water fund. Despite extensive lobbying with state and federal government agencies, none of these mechanisms were feasible.

In recent years, the watershed committee has functioned as an umbrella for the business and social sectors and communities that carry out reforestation actions every year in the upper watershed. Given the lack of state and municipal budgets for the

committee's operation, the CSO provides the counterpart budget to keep the watershed committee in operation. Governmental disinterest also manifested itself in the possibility of implementing a payment for environmental services program. The social difficulty was also present, since the payment for domestic water is minimal in this city, making it impossible to convince citizens to pay extra to improve watershed conditions.

(b)    Promoting Transparency and Accountability

In Mexico, municipal water utilities had limited technical and administrative capacity due to insufficient financial resources and the lack of skilled staff) [22].

### 4.2.4. City of La Paz (South Baja California)

In the city of La Paz, the population has a critical perception about water management. Among those surveyed in 2012, 45.3% considered that the management of the municipal water utility ranged from regular to bad; 74.4% considered that the water supply is limited to bad; 78.2% mention the bad taste of the water, and 93.4% of those surveyed did not use the distributed water for cooking [45]. This perception is justified given that the municipal water utility does not provide water quality data, or its distribution policy.

In 2013, several CSOs, universities and entrepreneurs formed the Citizen Observatory of Water and Sanitation to transparent water management. After more than a decade working with government agencies, especially with the municipal water utility, the CSO was invited in 2018 to join the Advisory Council of this utility as a representative of domestic users. From that moment on, and to strengthen governance, the CSO managed to change the meetings of this council from being behind closed doors to being public, transmitting them through social networks. Subsequently, to inform citizens and generate transparency and accountability mechanisms, several CSOs requested public access information on water management, budgets and projects through the National Transparency Platform, processing it for public understanding. These results are currently presented with the water utility in the so-called "Transparency Fairs" (http://elaguaenlapaz.mx/feria-de-transparencia-2019/ (accessed on 16 September 2020); https://www.comovamoslapaz.org/semana-del-agua-2da-feria-de-transparencia/ (accessed on 23 September 2020)

### 4.2.5. City of San Miguel de Allende (Guanajuato)

Another way to promote transparency in urban water quality is exemplified in the city of San Miguel de Allende. The CSO took up the studies elaborated by Ortega [26,50], who mentions the high concentrations of arsenic in wells, the main water source for domestic use, and acted to release this information, which the municipal water utility had never been reported. Contact and water consumption with excessive amounts of arsenic, a naturally occurring mineral, is known to cause skin, lung, urinary tract and kidney cancer, and other skin changes such as pigmentation and thickened skin (hyperkeratosis) [6].

Concerned about the opacity of water management by the municipal water utility, several CSOs formed a Citizen Water and Sanitation Observatory. Among their activities, they sought to inform the population about the water quality situation and to urge the municipality to take action.

Since 2012, tests and analyses have been carried out on an ongoing basis to monitor water quality from rural wells and urban area taps throughout the watershed. This work is conducted closely with community groups and with the support of several national and international universities (Texas A&M University, University of Guanajuato, Kansas State University and Northern Illinois University).

This information was subsequently shared and discussed with government institutions, in both the water and health sectors. However, government institutions did not acknowledge the data and did not place the issue on the public agenda. The City of San Miguel de Allende has a strong real estate, tourism, car assembly, and agribusiness sectors that would be directly affected by the information being disclosed by the CSOs in their growth and export plans. Therefore, these sectors put pressure on the municipal government to maintain the opacity of water quality data.

Public dissemination of the data became the responsibility of the CSOs, which publishes them in forums or on web pages (http://aguavidasma.org/ (accessed on 10 August 2020). The data obtained are public and presented in interactive maps (https://caminosdeagua.org/es/mapa-calidad-agua#parte-superior (accessed on 10 August 2020).

Studies show that the wells with the highest arsenic concentrations are located in rural areas, where more than 100,000 families depend on groundwater for drinking water. As a result, several CSOs continue to sample water from these wells, share the data with the ejidos, and recommend and implement alternatives such as rainwater harvesting systems. This eco-technique is being supported by CSOs in rural areas, both in homes and in public spaces (schools).

In addition, discussion workshops on family health and watershed health (the "Healthy Watershed, Healthy Community" program) are being held to educate and train local promoters and water quality monitoring committees in the ejidos.

### 4.3. Participation of Cities in the Conservation of Hydrological Services of the Watershed

Since 2003, Mexico has become home to one of the world´s most significant efforts to establish programs making payments for hydrological services (PHS) at the national level. In 2008, CONAFOR (National Forestry Commission) created a matching funds Program of Payment for Hydrological Services at the local level, seeking to incentivize cities to take a more active role in the management of the watersheds that provide them with hydrological services. This local program required at least 50% of program financing through local sources [56].

The matching programs also support ecological restoration, hydrologic services monitoring, and both cash and in-kind contributions for sustainable land-use practices, and gives local program operators greater autonomy over deciding whom and where to pay and how much [6,54]. This instrument was explored from the surveys investigating initial perception, conducted by various CSOs on willingness to donate.

#### 4.3.1. City of Xalapa (Veracruz)

From the survey conducted by the CSO in 2015, 30% of respondents have thought about supporting somehow the areas providing the water they consume. While 95% are not aware of the actions being carried out in the watershed, 77% consider it essential to create a fund to conserve the natural areas that supply water to the city, and 52% would be willing to pay an additional value in the water bill to protect the forest [53].

As shown in Figure 7, achieving a PHS with voluntary citizen input in the City of Xalapa took about 15 years. In that time, pressure from CSOs, academia and communities, and the attempt to coordinate with municipal and state authorities, allowed the development of various instruments and institutions that made it possible to implement the local PHS in 2020. Decision making and governance of the local PHS program, including verification and ensuring compliance with contracts, were quickly taken up by the existing committee.

This process was not free of challenges and obstacles, for which the citizenship information campaigns allowed CSOs to maintain a constant pressure. Currently, with the PHS underway, the ongoing challenge will be to make the work of the municipal water utility transparent in terms of quotas, projects and use of economic resources.

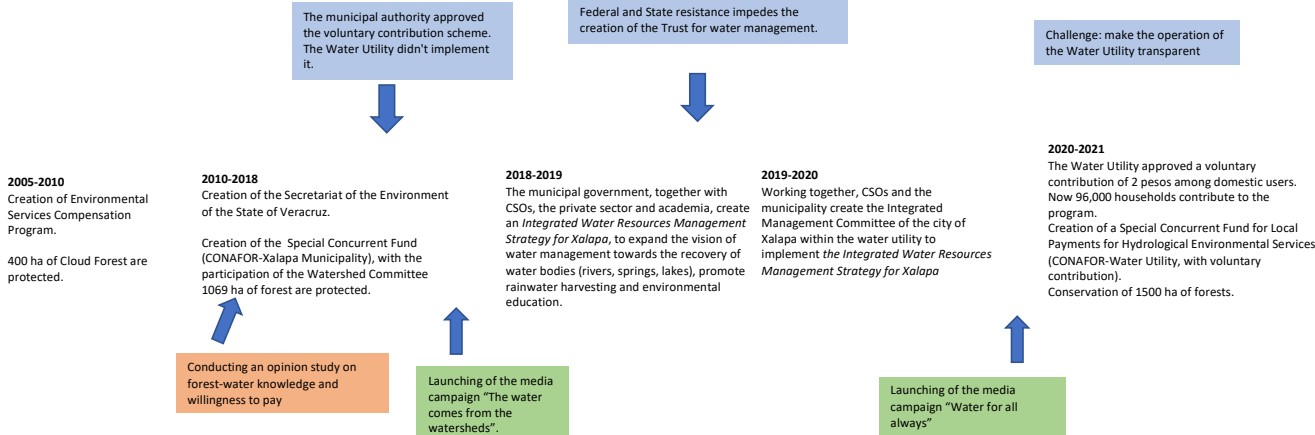

**Figure 7.** Formalization and Institutionalization of the Citizen Participation Process to Conserve of the Pixquiac Watershed (Xalapa, Veracruz).

### 4.3.2. City of Saltillo (Coahuila)

The City of Saltillo is located in an area of high water stress. For two decades (1980–2000) the city suffered a period of drought, which, together with poor water management, forced the municipal water utility to implement heavy rationing schemes. The municipal water utility's management was transformed and received heavy private investment in response to this situation. Surveys implemented by the CSO showed the population's willingness to pay voluntary donations to conserve the Protected Natural Area (PNA) of the Sierra de Zapalinamé, which supplies 30% of the city's wells. Since 1996, this PNA has been under the management of the CSO [52]. The process to reach a voluntary payment program to conserve Sierra de Zapalinamé strongly involved the municipal governments, the municipal water utility, the state government and the CSO, which was recognized from the beginning by the government authorities to create the management plan for PNA (Figure 8). Subsequently, this same CSO is chosen to receive voluntary contributions from businesses, households and government, to carry out restoration and conservation actions in the Sierra de Zapalinamé. In 2019 most of the donations came from residents of Saltillo. Accountability to donors is through the CSO's financial reports (Figure 9).

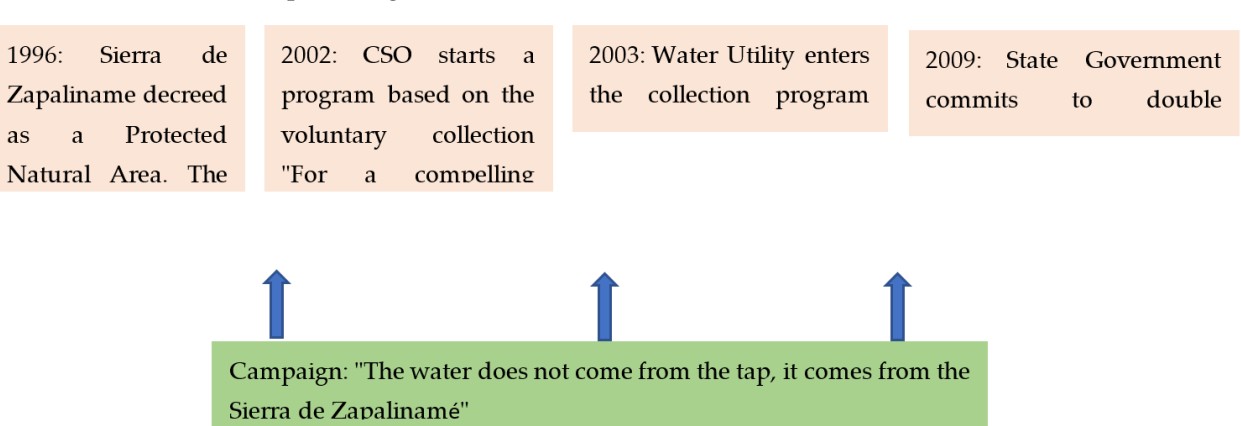

**Figure 8.** Process to Institutionalize Citizen Participation for the Conservation of the Sierra de Zapalinamé [52].

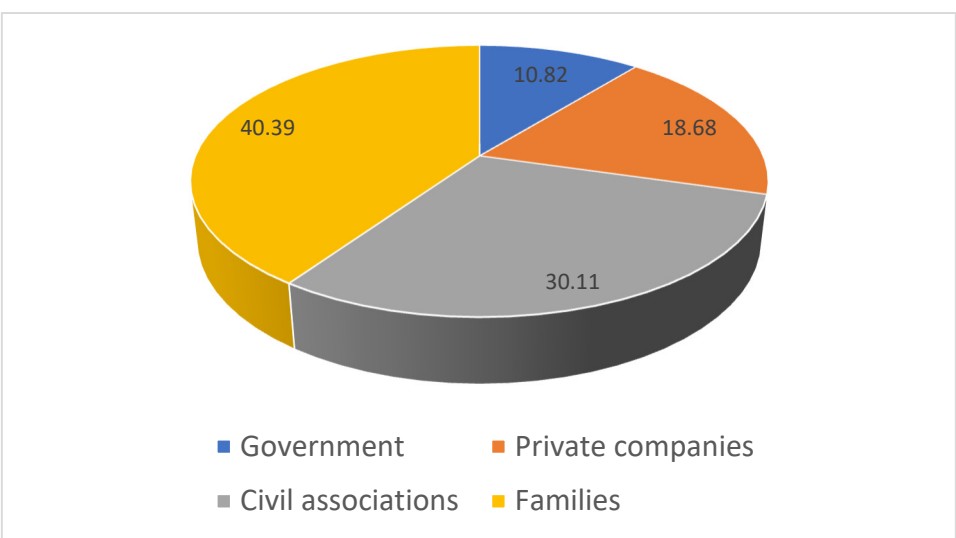

**Figure 9.** Voluntary Contribution (%) in 2019 for Sierra de Zapalinamé Conservation Program (www.zapaliname.org/informe-2019, accessed on 10 September 2020).

The budget collected is used for different activities, some chosen by the communities and others proposed based on studies conducted by the CSO. Some of the activities implemented with these resources have been to form and equip fire departments, reforest, promote productive projects in the communities of the Sierra, carry out education and outreach projects, conduct research and monitoring of local biodiversity and water quality, as well as equip the CSO with the necessary material for its work.

### 4.3.3. City of San Miguel de Allende (Guanajuato)

In 2007, in the municipality of San Miguel de Allende, an alliance of CSOs proposed and succeeded in formalizing a "Green Fund", which sought to allocate vehicle and environmental fines to a fund that would enable conservation actions in rural areas. This fund was advised by a technical committee formed by the CSO and academics.

Since then, until the installment of municipal government in 2019, municipal authorities did not maintain or respect the guidelines of the Green Fund, using its resources instead for urban infrastructure or other uses. Generally, the municipal governments of the City do not pay any attention to the rural area on which they depend for ecosystem services.

Recently, at the insistence of the CSOs with the current municipal government, the Green Fund and its Technical Committee have been recovered. Today, the Fund's budget is directed towards the rural areas where the CSOs work, coordinating fieldwork and technical support.

The erratic and fragile process of this fund responds both to the disinterest of municipal governments and conflicting interests with the business sectors and the difficulty of collaboration between CSOs in the area.

### 5. Discussion

Watershed-cities are representative of various climatic zones, but share common socio-environmental problems, such as water pollution and scarcity, deterioration of ecosystems, opacity and inefficiency of government institutions, which have prompted CSOs to build new instruments and institutions to strengthen governance, both in rural areas and cities, under a watershed approach. This effort took a long time and was not without obstacles from the government, business, and society.

The experiences described above indicate different lessons and challenges that must be met in the process of incorporating the cities into the management of watersheds in order to construct hydric resilience.

The first lesson is that for understanding the hydrological environmental services that the city receives, it is necessary to visualize and analyze it in a hydrographic unit, which can be flexible according to the primary water sources.

The second lesson is the scant knowledge of the urban population has on the origins of the water in their city and as an environmental service generated by watershed ecosystems. These perceptions have been generated as a response to traditional hydraulic management, based on grey infrastructure, which has characterized water management in Mexican cities [57]. Since urban residents often consider the quantity and quality of the water to be a theme exclusive to treatment plants, it is difficult for their vision to include the role of ecosystems as primary providers of water.

In cities dominated by technology and infrastructure, the conception of a society increasingly disconnected and independent of the ecosystems has been fomented [35,58,59]. This phenomenon has caused an urban blindness about the importance of maintaining the health of the watershed to provide ecosystem services. For this reason, environmental education concerning the watershed´s provision of hydrological ecosystem services was a necessary action in all of the cities. Environmental education seeks to develop an aware and informed public, with the capacity to assume commitments, to participate in the resolution of problems and to make decisions, and act to promote sustainable development [60,61].

However, current communication strategies need to be improved based on an analysis of the role of the actors and their degree of influence to define communication instruments more assertively [62].

The third lesson is related to the strengthening of governance. Since the governance of the watershed territory is currently fragmented between institutions at different scales, which divide the rural and urban spheres, the CSOs have created and strengthened agencies and instruments of governance that incorporate cities into watershed management, such as Watershed Committees, Citizen Observatories for Water and Consultative Councils.

In the face of the lack of transparency and poor representation in governmental agencies responsible for the management of water [55,63], CSOs have constructed new institutions through citizens alliances in order to fill gaps in the information and provide transparency, governance and proposals for alternatives, as in other regions [15,33]. The construction of an informed and active public enables reflection and feedback that sustains watershed management adaptation [64].

The formation of autonomous agencies, with diverse social representation, that monitor water quality in the cities as a counterpart of the government are vital due to the lack of transparency, mistrust and the poor capacity of the government agencies in terms of generating reliable information concerning the quality of water for human consumption.

Moreover, effective commons governance is easier to achieve when the resources and use of the resources by humans can be monitored, and the information can be verified and understood at relatively low cost [65].

The function of the CSOs, in terms of providing spaces of collaboration, enabling access to funding, generating information, conducting technical monitoring and maintaining a critical level of personnel [30] was fundamental to the strengthening of governance and transparency. This has gradually translated into greater informed and active citizen participation.

The fourth lesson lies in developing mechanisms for incorporating cities into watershed management. In two watershed cities (Xalapa and Saltillo), the local Payment for Ecosystem Services was achieved after many years. The differences in the elaboration of these programs show the importance of the political will of the municipalities and the flexibility of the operating agency since payments are public costs that can be put at risk by overall budget reductions and shifts in priorities across political regimes [66]. This same effort was frustrated in San Miguel de Allende and San Cristobal de las Casas, due to the disinterest of the municipal authorities and the low capacity to pay for water, conditions that are indispensable for the implementation of this type of program.

These instruments are not panaceas and still present broad challenges because there is still a shortage of evidence regarding whether or not these schemes improve quality of life

and generate desired behavioral changes. There is a mismatch between payment amounts and landowner opportunity costs [67]. Moreover, we still need to understand the complex relationship between forests and hydrologic services [56].

As Pfaff et al. [66] mentioned, some of the earliest and most successful adopters in the Matching Funds program are located in sites with some prior upstream-downstream success in coordination, where external intervention could be welcome as a tool to coordinate. This external intervention was in the form of CSOs in our cases studies.

Our results suggest that CSOs form alliances that strengthen collaborations among stakeholders at different scales, increasing government transparency and accountability, and providing a bridge of trust between upstream and downstream users.

## 6. Conclusions

The cases analyzed are representative of four climates of Mexico: hot arid, arid, humid subtropical and tropical. In all of them, the watershed´s contribution to the city water supply varies between 38% to 100%. However, the severe socio-environmental problems of contamination, scarcity, opacity in water management, and deterioration of ecosystems in the watershed are driving the organization and construction of an alternative, flexible governance built from the bottom up under the leadership of civil society organizations.

Each watershed analyzed is sui géneris in a physical, historical, political, cultural and leadership context. For this reason, the paths of each CSO have followed the same general route but with different actions, times and priorities.

The process of incorporating cities into watershed management is a long one. From the cases studied we can affirm that the construction of these institutions and instruments depends on several factors: (i) the political will of the officials of the municipal water utility and the municipality (for example Pixquiac watershed) (ii) the inertia of government institutions that do not promote dialogue and their reluctance to make data transparent (Laja river in Guanajuato and La Paz watershed), (iii) the pressures of the business sector (Laja river in Guanajuato), (iv) the history of political clientelism in rural areas that explain why the population is mobilized especially when economic resources are available (Pixquiac watershed), (v) the degree of external pressures in the watershed that cause socio-environmental conflicts (mining activities, infrastructure construction, real estate development), which the CSOs must address to maintain the integrity of the territory and which divert them from their plans and projects within the watershed (Laja river, La Paz, Mazatlán city, Pixquiac watershed).

Other challenges are more related to the CSOs themselves: (i) their dependence on external financing increases the vulnerability of these organizations, which means their priorities, plans and actions can vary over time; (ii) the actions they carry out in the territory are rarely subjected to an environmental impact of the actions that can be related to the integrity of the watershed and its cost-benefit; and (iii) the technical capacities of these organizations are still limited, and they are generally overwhelmed by the multiple threats to the territory in the form of extractive activities, constructions, megaprojects and security.

However, it should be clarified that the CSOs cannot replace the role of governments, whether local, state or federal. The importance of their participation lies in promoting transparency of actions, giving legitimacy to projects, working more closely and permanently with the local population, and forming a counterweight to governmental decisions.

Nevertheless, as Platt [3] stated, it is clear that sustainable watershed management is both an art and a science, and requires the participation of the public and of organized social actors. In this context, it is crucial to recognize the strategic achievements made by the CSOs in assessing and addressing persistent barriers and identifying challenges to consolidate their role in the management and governance of the watershed, incorporation of urban areas and financial sustainability of their participation.

**Author Contributions:** Conceptualization, H.C. and M.L.C.; methodology, H.C. and M.L.C.; valida­tion, H.C., M.L.C. and R.L.; formal analysis, H.C. and M.L.C.; investigation, H.C.; resources, J.M.F. and R.L.; writing—original draft preparation, H.C.; writing—review and editing, H.C., M.L.C. and R.L.; visualization, M.L.C.; supervision, H.C.; funding acquisition, R.L. and J.M.F. All authors have read and agreed to the published version of the manuscript.

**Funding:** This research received no external funding.

**Institutional Review Board Statement:** Not applicable.

**Informed Consent Statement:** Informed consent was obtained from all subjects.

**Data Availability Statement:** Not applicable.

**Acknowledgments:** This paper seeks to show the process, actions and results of the CSOs to strengthen the environmental governance in seven watersheds of Mexico. Behind each of them are professionals, technicians and volunteers who make these changes possible, and who work every day in favor of better watershed management. We would like to thank all of them for their trust in sharing their experiences with us. We thank the Fondo Mexicano para la Conservación de la Naturaleza for their support in managing the meetings with the CSOs. The authors would like to thank the reviewers for their comments and suggestions. Their comments definitely improved this article.

**Conflicts of Interest:** The authors declare no conflict of interest.

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
