# Peer review of "Environmental Governance in Urban Watersheds: The Role of Civil Society Organizations in Mexico"

_sustainability, doi:10.3390/su14020988_

Round 1
Reviewer 1 Report
The paper deals with an arguably salient and under-researched topic. The general structure of the paper is easy to follow, the different chapters are well balanced in contents and length. This being said, I have however some major remarks regarding the paper in its present form.
Introduction
While the paper may be addressing an exciting research problem, it fails to set the wider context and the debates that the research is contributing to. You do not highlight the main field of literature to address the research questions and you do not clearly identify a tension in existing understandings that the research will attempt to resolve. How other scholars and academicians have analysed this issue? What research methods have been used in literature?
Moreover, you do not clearly explain what is new, fresh and original about this research and do not show how it adds to the existing literature.
Methodology
In the section “Methodology”, you provide a list of “qualitative sources”. You do not explain how the data was gathered and analysed. What methods do you use to analyse data from “qualitative sources”? Have other scholars and academicians use this method? Why do you use this method? How do you gather data?
Reviewer 2 Report
The article covers a relevant and current topic within the field of research in the governance of river areas: the scale of governance and specifically the urban-rural linkages. In particular, the article’s aim is to analyse the role of Civil Society Organizations (CSO) in environmental governance at a city level. The main contribution is to focus the study on the role of CSOs in such a process of watershed governance at a city level. The selected case studies (7 CSO in Mexico in which watershed-city relationships have been developed) are rich enough to generate a great discussion. However, the information is not well-structured in the different sections composing the paper (introduction, methodology, results, discussion and conclusion).
In this sense I recommend to structure the introduction in two parts. A first part on: 1) ecosystem services related to watershed (few references are mentioned and more recent articles should be included, see line 51), 2) socio-environmental problems associated, and 3) the lack of multiscale and transversal governance. And a second part on the role of CSO in urban environmental governance. Both parts represent the background and the research aim and research questions (as it is a qualitative research we normally refer to research questions) should be announced at the end of this two parts that compose the Introduction. In particular, after contextualizing the case of CSO and the Water and Cities Program (lines 133-144).
Regarding the Methodology section, it should include 3 subsections: 1) the case study, 2) the data collection process, 3) the data analysis process. Normally before starting detailing each section, an introductory paragraph arguing the qualitative nature of the approach is suitable (why qualitative research is best situated to answer the research questions). In relation to the case study, it should incorporate figure 1 and table 1 together with the written explanation. However, table 1 should not present results, this is “main problems in the watershed/sub-watershed”. The sentence in lines 165-168 needs revision. It is not clear and connected. In relation to the data collection, information is needed to understand when the data collection was conducted and a summary table presenting the sample would help (i.e. number and person interviewed per watershed, title and number of documents consulted per watershed, type of actions reviewed in the visits...). In relation to data analysis, any explanation is made. Please explain how data was processed (transcriptions, content analysis, codification process, triangulation, reliability, etc).
Regarding the Results section, a summary table synthetizing the CSO objectives/priorities and structure in terms of participants, the socio-environmental problems identified, the initiatives/campaigns developed, the alliances propelled would be of great interest to then facilitate the reading of the main topics developed in the results section. Regarding the first identified theme “Mechanisms for Incorporating Cities into Watershed Management”, 3 subthemes are explained: 1) hydrographic unit delimitation, 2) population surveys, 3) awareness-raising campaigns. Now 2 and 3 are presented together and I think they should be explained separately although they are interrelated. In the second identified theme “Construction of Governance in the Watershed and Water Management”, no literature references should appear as this section present results of your research. You can then discuss about them and contrast with the literature in the discussion section. Figure 5 is really nice and is then well developed in the text the parts referring to the new forms of governance. However, the left and central information presented in the figure on key actors and urban-rural linkages is not presented anywhere and would be of great interest. Maybe at the beginning of the results section, after the suggested summary results table comparing the 7 case studies.
The sentence in lines 175-176 is correct but does not present the subtheme announced “promoting transparency and accountability. It would be nice to add a general presentation of this theme before detailing the selected case studies.
Regarding the last theme identified, “Participation of Cities in the Conservation of Hydrological Services of the Watershed”, maybe the title should refer to funding types as it is the main topic developed (voluntary donations or environmental fines, depending on the case study).
Discussion and conclusions need further revision. Especially since some interesting and relevant topic are first introduced in the conclusion and should therefore appear before in the results as descriptions and in the discussion as interpretations. I refer to the topics listed in lines 404-417. Conclusions should be consistent with the evidence and arguments presented before.
Finally, ethics statements and data availability statements are not referred to and it would be nice to mention it.
Recommended references:
Water governance:
Ida Widianingsih, Caroline Paskarina, Riswanda Riswanda & Prakoso Bhairawa Putera (2021) Evolutionary Study of Watershed Governance Research: A Bibliometric Analysis, Science & Technology Libraries, DOI: 10.1080/0194262X.2021.1926401
Ecosystem services:
MEA - Millennium Ecosystem Assessment. (2005). Ecosystems and human well-being.Washington, DC: Island Press.
Antonio J. Castro,Caryn C. Vaughn,Jason P. Julian,Marina García-Llorente (2016). Social Demand for Ecosystem Services and Implications for Watershed Management. JAWRA-14-0170-P, https://doi.org/10.1111/1752-1688.12379
Round 2
Reviewer 1 Report
The manuscript has been revised according to the suggestions and comments of the reviewer
Reviewer 2 Report
Thanks for your revision. I consider the manuscript is now well structured and clear as well as interesting.